# A Qualitative Study of Rural Plant-Based Eaters’ Knowledge and Practices for Nutritional Adequacy

**DOI:** 10.3390/nu16203504

**Published:** 2024-10-16

**Authors:** Michelle Leonetti, Jane Kolodinsky, Amy Trubek, Emily H. Belarmino

**Affiliations:** 1Food Systems Program, University of Vermont, Marsh Life Sciences, 109 Carrigan Drive, Burlington, VT 05405, USA; michelle.leonetti@uvm.edu (M.L.); jane.kolodinsky@uvm.edu (J.K.); atrubek@uvm.edu (A.T.); 2Department of Community Development and Applied Economics, University of Vermont, Morrill Hall, 146 University Place, Burlington, VT 05405, USA; 3Gund Institute for Environment, University of Vermont, 617 Main Street, Burlington, VT 05405, USA; 4Department of Nutrition and Food Sciences, University of Vermont, Marsh Life Sciences, 150 Carrigan Drive, Burlington, VT 05405, USA

**Keywords:** vegan, vegetarian, flexitarian, nutrition knowledge, healthy diet, at-risk population

## Abstract

(1) Background: Healthful plant-based diets, especially those rich in minimally processed plant-based foods such as fruits, vegetables, and whole grains, have been associated with a lower risk of diet-related chronic disease. However, individuals who limit or avoid animal products may be at risk of nutrient deficiencies, especially related to Vitamin B12, Vitamin D, omega-3 fatty acids, calcium, iron, iodine, zinc, and protein. Such deficiencies can result in both short- and long-term health challenges. We used qualitative methods to explore plant-based eaters’ knowledge and awareness of these eight nutrients of concern in diets that limit or exclude animal-source foods. (2) Methods: We conducted interviews with 28 rural flexitarian, pescatarian, vegan, or vegetarian adults in Vermont, USA. (3) Results: The participants positively viewed the healthfulness of plant-based diets, but many recognized limitations to accessing Vitamin B12, Vitamin D, and omega-3 fatty acids. They shared the strategies used to meet their needs including eating a varied diet, seeking out specific foods, and taking dietary supplements. Analyses identified gaps in the participants’ knowledge related to nutrient bioavailability, food sources of specific nutrients, and the importance of zinc and iodine. Vegans—the highest-risk group—generally presented as the most knowledgeable. The participants noted a lack of local plant-based nutrition expertise. (4) Conclusions: Addressing the identified knowledge gaps and challenges to dietary adequacy, especially among those who limit, but do not fully omit, animal-source foods, may support plant-based nutrition.

## 1. Introduction

Plant-based diets are increasingly promoted for their potential to support health and sustainability goals [1,2]. The retail sales of plant-based foods grew 44.5% in the United States (U.S.) from 2019 to 2022, with a market value of USD 8 billion [3], and over one quarter of Americans (28%) report reducing their meat intake [4,5]. The term “plant-based diet” includes a wide range of dietary patterns that include larger amounts of foods derived from plants such as vegetables, fruits, grains, legumes, nuts, and seeds and lower amounts of animal-source foods [6,7]. While this includes vegetarian (excludes meat, 2% of U.S. adults) and vegan (excludes all animal-source products, 2% of U.S. adults) diets, it also encompasses less restrictive dietary patterns that include animal products and byproducts such as pescatarian (includes fish, 2% of U.S. adults) and flexitarian (omnivore but with limited meat consumption, 22% of U.S. adults) [5].

Mounting evidence suggests that plant-based diets high in minimally processed foods such as fruits, vegetables, and whole grains are associated with a reduced risk of diet-related chronic diseases like type 2 diabetes, hypertension, and cardiovascular disease [8,9,10,11,12] and with improved digestion and gut microbiome diversity due to their dietary fiber and phytonutrient content [13,14], whereas diets high in animal products—especially red and processed meats—are linked to adverse health outcomes including type 2 diabetes, cardiovascular disease, and certain cancers due to their high cholesterol and saturated fat levels [15,16].

While minimally processed plant-based diets have been associated with positive health outcomes, some micronutrients are either not present in sufficient quantities in plant-based foods or are not as easily absorbed. For example, compared to omnivores, vegans and vegetarians are at risk of deficiencies of some nutrients, specifically Vitamin B12, Vitamin D, omega-3 fatty acids, calcium, iron, iodine, zinc, and protein [17,18]. Table 1 highlights the key nutrients of concern, dietary sources, key roles in the body, and potential limitations for people eating a plant-based diet. To address these nutritional gaps, experts recommend careful dietary planning that includes consuming a variety of plant-based foods to cover a wider nutritional base in conjunction with fortified foods, vitamin/mineral supplements, and foods with specific nutrients like nutritional yeast, algae, and chia seeds [18]. Understanding to what extent plant-based eaters are aware of and seek out these nutrients is essential for assessing the overall healthfulness and sustainability of the dietary pattern.

Although meat consumption is common in rural areas [19], many rural residents are already consuming diets that may be categorized as “plant-based”. For example, a recent statewide survey in the state of Vermont found over a third of the adult population to be limiting or avoiding meat intake [20]. However, rural plant-based eaters may face greater barriers to consuming a nutritionally adequate diet. Despite rural areas growing most of the food produced in the United States, access to and consumption of a diverse selection of plant-based foods may be limited due to the long distances to stores, limited stocking, or financial barriers [21,22]. Further, cultural norms and expectations to eat meat [19,23,24] could create dietary challenges for rural residents who choose to limit or exclude meat from their diet.

In this study, semi-structured interviews were conducted with rural plant-based eaters who believed themselves to have healthy diets. The objectives were to determine the level of knowledge and awareness of the nutrients of concern among plant-based eaters and identify where they access nutrition information. With the number of plant-based eaters set to continue to increase in the future [3,4,5], information on knowledge areas and knowledge gaps can be used to guide public health action and healthcare practice in rural communities.

## 2. Materials and Methods

### 2.1. Study Participants and Data Collection

The target population for this study was rural residents in the U.S. state of Vermont who eat a high-quality plant-based diet (i.e., positive deviants with respect to dietary intake). We defined “high quality” as self-identifying as a healthy eater and meeting the national recommendations for fruit and vegetable intake. We considered a person to have a “plant-based” diet if they limited their intake of red meat, poultry, or fish to no more than four times per week. In light of the strong meat-eating attitudes and expectations for daily meat consumption in rural areas [19,24], this new approach to defining a plant-based diet included individuals who consume meat products most days, but not all.

The participants were recruited through community organizations, local food co-ops, plant-based restaurants, and online forums by asking them to display physical posters, include announcements in their newsletters, or post the recruitment message to their social media pages. The recruitment materials targeted individuals with vegan, vegetarian, semi-vegetarian, and/or flexitarian diets. Any person who saw a recruitment advertisement could take an online survey to determine their eligibility based on their age being ≥18 years; residence in rural Vermont; self-identification as a healthy eater; and reporting typically eating meat less than five times per week, at least 2.5 cups of vegetables per day, and at least 2 cups of fruit per day based on evaluative questions. The residents in all the counties of Vermont were eligible except for those in Chittenden County due to the substantially larger population and population density in Chittenden County as compared to the state’s other counties [25]. The geography was verified using geolocation. To support sample diversity, recruitment quotas were set at: (1) ≤6 people per county, (2) ≤20 people over or under the age of 50 years and (3) ≤22 people of the same gender. Of the 480 people who completed the screening survey, 76 met the inclusion criteria and were evaluated for inclusion against the geolocation data and the demographic quotas. Within each quota, eligible individuals were contacted in chronological order (until the quota was filled) and invited to take part in an interview. Forty people were excluded because their geolocation confirmed that they were not located in Vermont and three because the recruitment quotas were full for at least one of their demographic characteristics by the time that they completed the screening survey. Five people did not respond to our request to conduct an interview.

Prior to the interview, the subjects completed a brief online demographic and diet survey including questions about the specific nutrients of concern in a plant-based diet (Appendix A). The interviews were conducted via phone or videoconference and ranged between 24 and 69 min in length. The participants were compensated USD 25. The study protocol was determined to be exempt by the Institutional Review Board at the University of Vermont (protocol ID# 00002156).

Building from prior studies that successfully applied the Theory of Planned Behavior (TPB) to understand and predict plant-based eating [6,26], a semi-structured interview guide (Appendix A) was developed based on the core concepts of the TPB with an additional section that sought to capture the knowledge and awareness of the nutrients of concern in a plant-based diet. The questions explored overall diet, motivations for engaging in a plant-based diet, ease of engaging in a plant-based diet, family/community views, suggestions for what could make eating a plant-based diet easier for others, and the degree of nutritional planning for the specific nutrients of concern in a plant-based diet: Vitamin B12, Vitamin D, omega-3 fatty acids, calcium, iron, iodine, zinc, and protein. Each participant was asked tailored questions about the nutrients of concern based on how they responded to the corresponding pre-interview survey questions.

### 2.2. Analysis

The interviews were audio-recorded, transcribed, manually cleaned, imported into NVivo qualitative data analysis software (QSR International Pty Ltd., Burlington, MA, USA, Version 20 Release 1.6.1), and structurally coded by question. A template coding and analysis approach [27] was used to capture the strategies and contextual factors that may explain the ability to adopt and maintain a plant-based diet, and a deductive approach informed by Grounded Theory [28] was used to examine the degree of nutrition planning related to the nutrients of concern commonly lacking in plant-based diets. There were 11 interviews openly coded by the first author, and two research assistants met to discuss emergent ideas and themes. Based on these discussions, two codebooks (“Adoption and Maintenance” and “Nutrients of Concern”) were created and iteratively refined before being used by the main author and one of the research assistants to code all the interviews. The focus of this analysis is the data that were coded in the “Nutrients of Concern” codebook which focused on the knowledge and degree of nutritional planning (Appendix A). Prior to coding all the interviews by using the final codebook, the first author and a research assistant—both with expertise in nutrition—achieved a “substantial” level of agreement determined by an inter-rater reliability of kappa >0.61 (kappa = 0.69 and prevalence-adjusted and bias-adjusted kappa = 0.98) [29] for two interviews. The first author wrote a memo for each code to synthesize the key themes. Prior to finalization, the memos were reviewed, discussed, and refined in collaboration with the research assistant.

The data were further analyzed by taking in-depth notes on each participant and identifying the key characteristics through both the interviews and pre-interview survey responses, which allowed them to be understood contextually. The participants were grouped and characterized by diet type and other similarities such as the level of nutrition knowledge, geography, and specific ideologies/practices. Through this process, the key quotes were identified and incorporated into illustrative examples to demonstrate significant ideas.

## 3. Results

Twenty-eight adults ranging from 19 to 77 years of age participated in the interviews (Table 2). Most (88%) of those who disclosed their race identified as non-Hispanic white, similar to the demographic composition of rural Vermont [25]. Three-quarters (75%) of the participants were female. All self-identified as having average or above average nutrition knowledge. There was diversity in diet type, but over 1/3 (*n* = 11) self-identified as vegan. Most (82%) reported eating a plant-based diet for 10 years or longer, including two participants who were raised plant based since birth and several who had been eating plant based for 40+ years.

### 3.1. Overall Attitudes about the Healthfulness of Plant-Based Diets

The participants viewed plant-based diets favorably due to health, environmental, and ethical considerations. Overall, the participants believed that a plant-based diet is healthful: “I’ve seen it work successfully [for myself]… and lots of people in my life” (Vegetarian, P#06). In fact, about one-third of the participants (*n* = 10) stated that they did not have any significant concerns about the nutritional adequacy of a plant-based diet. For example, one pescatarian shared, “There’s not much you can’t get [on a plant-based diet]… and I don’t have any particular concerns” (P#09). However, there was an appreciation that plant-based eaters need to be “conscious” or “aware” of a few key nutrients, including Vitamin B12, omega-3 fatty acids, iron, protein, Vitamin D, calcium, iodine, and zinc in descending order of predominance. Despite this, there was a general belief that it is easy to meet the nutritional needs on a plant-based diet through dietary planning, supplementation, and a varied, minimally processed diet.

A common sentiment was that a plant-based diet is more nutritious than a meat-centered diet and can positively impact overall health and wellbeing. As one 77-year-old vegetarian explained, “At my age… I keep expecting [my health] to get worse… but I might actually be getting better in some areas [due to my diet]” (P#18). Several noted that they were significantly healthier than their peers and credited a plant-based diet with a positive impact on chronic health conditions (especially cholesterol and cardiovascular health), aging, and energy levels. The participants also shared the belief that eating a plant-based diet helps to reduce the risk of ill health, for example, “inflammation, your risk of cancer, your risk of diabetes, basically all chronic illnesses” (Vegan, P#25).

The participants put emphasis on the consumption of a varied diet to “cast a wide net”, as opposed to targeting specific nutrients. About half of the participants (*n* = 14) mentioned eating a varied diet to ensure nutritional adequacy: “[My family tries] to have a fairly varied diet on a regular basis, so eventually you’re gonna get everything you need” (Pescatarian, P#09). The participants felt that by doing this, some of the mental labor needed to plan was reduced. For example, one flexitarian shared, “I don’t really think about it that much anymore… I just kinda do it”. Other examples included focusing on balance, macronutrients, local produce, and varied cuisines, with or without the use of dietary supplements. Although not universally used, numerous participants consumed multivitamins to cover potential nutritional gaps or sought out specific nutrients that they believed may be lacking in their diet.

There was also a distinction made between a processed plant-based diet and a whole-food plant-based diet. Close to half of the participants (*n* = 13) discussed the idea that a plant-based diet is not necessarily healthy because processed vegan food is readily available. As one vegan put it, “Vegan doesn’t equal healthy… there’s so many like junk food vegans and like so many things that like are completely unhealthy and still vegan” (P#24). Several others shared this sentiment stating that it was easy to overconsume processed foods, thus making it more difficult to obtain the nutrients and fiber that would be found in whole foods. However, a subset expressed that a vegan diet that includes substantial quantities of processed foods still may be “just a little bit healthier” than the standard American diet and could serve as a transition food or for convenience.

### 3.2. Key Nutrients of Concern

The following sections summarize the knowledge and awareness about each of the nutrients of concern for a plant-based diet. The key insights and illustrative quotes for each nutrient queried are presented in Table 3.

#### 3.2.1. Vitamin B12

The participants who expressed concern for Vitamin B12 were primarily vegans and vegetarians and one-third of the participants (*n* = 10) mentioned it in the context of a plant-based diet. As one vegan put it, “B12 is… a vitamin that you get basically from animal products, so being a vegetarian, I just don’t get that” (P#15). There was acknowledgement that Vitamin B12 is difficult to obtain from plant-based foods and for the most part must be supplemented. For two vegans, Vitamin B12 was the only nutrient that they were concerned about not obtaining from their diet.

Vitamin B12 was the mostly commonly supplemented nutrient discussed, with various forms noted including liquid sub-lingual supplements, pills, and multivitamins. Ten participants described specific food sources that they seek out for Vitamin B12. Over half (*n* = 16) mentioned using nutritional yeast (a food source high in Vitamin B12) for its cheese-like flavor. While six participants acknowledged the nutritional content of nutritional yeast as important, only three use it primarily for nutritional purposes and acknowledge that it “only goes so far”. Most of the participants who enjoy nutritional yeast and use it for nutritional content were vegans or vegetarians.

While nutritional yeast was the most common dietary source of Vitamin B12, fortified foods such as plant-based milk alternatives were mentioned by two participants. An additional 15 participants reported that they consume plant-based beverages but did not mention Vitamin B12 fortification. One pescatarian mentioned that they do not supplement Vitamin B12 because they eat eggs, but no other non-vegan participants noted seeking out meat or milk as a source of Vitamin B12.

#### 3.2.2. Vitamin D

Vitamin D was the second-most mentioned nutrient. The participants expressed the idea that it is challenging to obtain enough Vitamin D, especially in Vermont in the winter, i.e., “we don’t live in a sunshine state… and my own Vitamin D levels were low this past winter” (Flexitarian, P#01). Outside of geographical and seasonal considerations, the participants indicated that there are not many foods that have Vitamin D—the only specific example discussed was fortified plant-based milks which were only mentioned by one participant. There was no mention of cow’s milk or fish as sources of Vitamin D despite their inclusion in many plant-based dietary patterns.

As such, supplementation was the most common conscious source of Vitamin D for the participants. Vitamin D supplements were often taken on the advice of a medical professional. One participant shared that their doctor recommends that “every Vermonter” should be taking Vitamin D due to the northern location. No participant said that their healthcare provider recommended it specifically because of a plant-based diet.

#### 3.2.3. Omega-3 Fatty Acids

About one-third of the participants (*n* = 10) discussed omega-3 fatty acids unprompted, including nearly half of the vegans. While there was some awareness of the need to incorporate omega-3s into a plant-based diet, the respondents displayed limited knowledge regarding the differences between plant-based alpha-linolenic acid (ALA) and animal-based eicosapentaenoic acid (EPA)/docosahexaenoic acid (DHA). For example, only one vegan participant noted that ALA can be converted into DHA/EPA if consumed in sufficient amounts. Of the five participants who discussed taking omega-3 supplements, four were vegans who took algal-based supplements; the other was a pescatarian who took fish oil. While only three mentioned seeking out nuts/seeds specifically because they are “jam packed” with omega-3s, most of the participants reported enjoying the consumption of flax, chia, and/or hemp seeds. These seeds were used interchangeably as additions to foods like yogurt and smoothies and were commonly sought after as a good source of fiber and protein or to fill in general gaps in their diet. Notably walnuts, a rich source of ALAs, were mentioned less than other types of nuts that are lower in ALAs (e.g., almonds and cashews).

#### 3.2.4. Calcium

About 40% of the participants (*n* = 12) discussed calcium. Of those, three vegans felt that calcium is a nutrient that needs to be monitored on a plant-based diet, but all expressed that it is not difficult if you know the sources. However, four participants mentioned a general concern for calcium in the context of other health conditions such as bone health and aging. In these cases, the concern was not related to plant-based diets, but it was a driver for consuming more calcium-rich foods.

A single flexitarian with training in nutrition mentioned that the bioavailability of calcium in plant sources “can be really limiting” (P#01) as compared to animal sources, and a vegan accurately articulated that Vitamin D is needed to aid in calcium absorption. Neither of these points were expressed as common knowledge among the sample.

Spinach, dark green leafy vegetables, and cruciferous vegetables were discussed as sources of calcium by four participants. Meanwhile, four others discussed dairy as a source of calcium and were either concerned about obtaining enough because they do not consume dairy products or were not concerned about obtaining enough calcium because they do eat dairy products. Non-vegans were generally less concerned about calcium due to the dairy in their diet. All who were concerned about calcium in a plant-based diet were vegan or did not consume a significant amount of cow’s milk. However, three discussed fortified non-dairy milk alternatives as a source of calcium.

Overall, when the participants discussed calcium, the focus was on food sources; no participants mentioned seeking out calcium supplements. One pescatarian expressed that, “if I can get calcium from black beans and from bok choy [instead of fortification]…, I would prefer that”.

#### 3.2.5. Iron

Slightly less than half of the participants (*n* = 13) discussed iron. A few noted (*n* = 5) iron as a nutrient of concern in a plant-based diet, but more participants (*n* = 8) discussed iron in the context of a general deficiency or life stage (i.e., menstruation, menopause), and uncertainty was expressed about whether deficiencies were more likely related to a plant-based diet or a health condition.

Only three participants mentioned taking an iron supplement due to concerns of iron deficiency; otherwise, there was some ambivalence about iron: ss one flexitarian expressed, “I do wonder about iron, but I don’t wonder about it enough to like do anything about it” (P#14).

The only participants who noted the decreased bioavailability of iron in plant-based foods had nutrition training; no participants mentioned the difference between heme and non-heme iron or the use of ascorbic acid to increase absorption.

Overall, there was a focus on obtaining iron through a variety of plant-based foods like spinach, dark green leafy vegetables, and legumes as opposed to supplementation. However, one vegan mentioned that they likely obtain enough iron from vegetables but take a supplement occasionally just to be sure. Red meat was not emphasized as important for meeting iron needs, and no flexitarian participants mentioned regularly seeking out animal foods like red meat as a source.

#### 3.2.6. Iodine

Overall, iodine was not of significant concern for the participants in any diet type. Of the six who discussed iodine, only one mentioned it in connection to plant-based diets. Seaweed and iodized salt were the main dietary sources of iodine, and only two participants reported taking a supplement with iodine, both in the form of a multivitamin. Of the 13 participants who described consuming seaweed, about half (*n* = 7) mentioned the knowledge or use of seaweed for its nutritional qualities. However, three people mentioned that they do not enjoy the “ocean” flavor, and do not eat it despite the nutritional benefits. Seaweed was also widely consumed by the participants who did not mention the nutrition content. While most enjoyed seaweed as part of sushi, adding it to recipes was also popular. Sprinkling dulse flakes over foods was mentioned by five as an iodine source or a salt alternative. While ten participants discussed iodized salt, six participants mentioned using it infrequently, with a preference for Himalayan, sea, or kosher salt. Only two participants mentioned using iodized salt as a source of iodine.

#### 3.2.7. Zinc

Of the nutrients queried, zinc was the least mentioned, and no one raised concerns in the context of a plant-based diet, including decreased bioavailability. Although the participants often discussed eating foods naturally rich in zinc such as legumes, tofu, nuts, and seeds, none described seeking out specific foods as a source of the mineral. A few (*n* = 5) discussed the inclusion of zinc in their supplementation routine, especially as part of a multivitamin. Among these individuals, the focus was on the perceived benefits for aging and immunity.

#### 3.2.8. Protein

All 28 participants were asked to discuss protein, but many also commented on it unprompted. Few worried about whether plant-based diets have adequate protein; more commonly, the participants indicated that they believe that protein is given excessive attention in American culture: “I think we overemphasize protein as a culture, so I don’t ever worry about that and I feel strong… I’ve never felt weak… as a vegetarian” (Pescatarian, P#27). Notably, no vegans mentioned any concerns about consuming adequate protein.

About one-third of the participants (*n* = 9) discussed protein complementing. This dietary approach involves combining two or more incomplete protein foods to obtain all nine amino acids essential for the body. Vegans and vegetarians were the most knowledgeable, while flexitarians and pescatarians were either less knowledgeable or less concerned. Of the six participants who discussed pairing beans and grains to create a complete protein, two vegans accurately noted that complementing does not have to happen at the same meal. A subset of vegan participants also mentioned seeking out plant-based foods with complete protein such as tofu and quinoa, so that complementing is not necessary. Two pescatarians shared that they are aware of protein complementing but consume a varied diet to meet their amino acid needs instead of actively combining foods.

A wide variety of protein foods were mentioned. The most common plant-based sources were tofu, tempeh, and legumes. Of note, six participants reported beans as being a staple in their diet, consuming them daily or weekly. For non-vegans, animal-based protein sources such as fish, poultry, eggs, and dairy were important aspects of obtaining enough protein, but they also consumed a variety of plant-based proteins.

About one-fifth of the participants (*n* = 6) shared that they prefer whole-food protein options (such as beans, nuts, and seeds) over more processed protein foods. There were some negative health perceptions about heavily processed meat alternatives, but five participants expressed the idea that less processed convenience foods like precooked tofu made plant-based proteins more accessible.

### 3.3. Source of Nutrition Information

Over 80% (*n* = 23) of the participants discussed both formal and informal sources of nutrition information. Health practitioners were the most mentioned source (*n* = 12). Of those who mentioned health practitioners, most said that they obtain nutrition information from mainstream practitioners or bloodwork results. Among these participants, medical professionals were trusted sources, albeit perceived to have limited knowledge of nutrition, especially plant-based nutrition. Several participants mentioned that they wished that they had access to providers with training in plant-based nutrition and/or that plant-based nutrition programs (e.g., receiving produce vouchers from the doctor or cooking classes at health centers) were more widespread in their communities. There was limited reference to registered dietitians. Four participants mentioned receiving nutrition and supplementation advice from alternative health professionals such as naturopaths, acupuncturists, and herbalists. About one-third of the participants (*n* = 10) discussed receiving nutrition information through informal sources, such as discussions with friends and family and mixed media (e.g., books, podcasts, websites).

## 4. Discussion

To our knowledge, this study is the first to explore the knowledge and awareness of plant-based eaters in the rural U.S. with respect to the eight dietary factors identified by the Academy of Nutrition and Dietetics as the nutrients of concern in diets that limit or exclude animal-source foods [17]. We found that the respondents were largely unburdened by nutrition concerns. Most focused on meeting their nutritional needs using whole-food ingredients that are readily available from rural grocery retailers (e.g., fruits, vegetables, nuts, seeds, and legumes) or supplementation. While we intentionally recruited positive deviants with respect to plant-based eating in rural areas, we still found important gaps in their nutrition knowledge, planning, and local resources that could contribute to nutrient deficiencies. These insights can be used to inform nutritional guidance and interventions, as the number of plant-based eaters is expected to continue growing [4,5].

While few Americans fully exclude meat from their diets, over one-quarter report limiting their meat consumption [5]. The finding that those who do limit but do not go fully vegan/vegetarian have more substantial knowledge gaps is important for clinicians and public health professionals. While nutritional planning for the eight nutrients evaluated in this study is most critical for those with the most restrictive diets [18,30], other plant-based diets that are poorly planned may still fall short of these nutrients. Despite many participants eating plant-based diets for more than 10 years, they may still benefit from nutrition counseling, a service that may be difficult to access in some rural areas.

The participants were particularly aware of the need for Vitamin B12, Vitamin D, and omega-3s, which is important because these nutrients may be particularly challenging to access through foods, especially for vegans (Table 1). While nutritional yeast was used to obtain Vitamin B12, it is less effective at improving Vitamin B12 status than supplementation [31] and may not be accessible to all rural residents as it is unlikely to be a staple item for all grocery retailers. Similarly, fortified non-dairy beverages were used as sources of Vitamin B12 but fortification in plant-based beverages can vary significantly between brands [32] or be absent from homemade versions. There may also be limited absorption of nutrients in plant-based beverages either due to the form of fortification, lack of solubility, or binding with plant components [32,33].

While Vitamin D can be produced by the body, it is difficult to create in sufficient quantities in northern Vermont. Recommending cow’s milk, fortified plant-based beverages, and fatty fish can be helpful, but these products may be insufficient to meet nutritional needs in isolation [34]. Thus, supplementation, especially among those who do not consume fish and in the winter, would help ensure adequacy. However, among the participants in this study, Vitamin D was already the most recommended supplement by health practitioners suggesting that additional action or information around Vitamin D may not be a priority. This is supported by the data from the National Health and Nutrition Examination Survey (NHANES) which shows that Vitamin D is the most supplemented nutrient for all age groups 20–60+, aside from a multivitamin [35].

The participants demonstrated limited knowledge regarding the reduced availability of EPA/DHA omega-3s, suggesting a need for increased education, particularly for those who do not consume fish. Prior research has found the intake of EPA/DHA omega-3s to be low in vegetarians and extremely limited in vegans [36]. Despite those who do not eat fish expressing more knowledge and concern about omega-3s and the relative popularity of ALA-rich nuts and seeds, it is unlikely that this population is consuming sufficient ALA omega-3 fatty acids from plant sources due to the low and variable conversion rate [36]. A fish oil or algal supplement may be a helpful dietary complement but comes with cost considerations. While encouraging the intake of fortified foods like nutritional yeast, nuts, seeds, and fortified beverages may be helpful, and supplementation and routine monitoring may offer more effective means of ensuring the adequacy of these nutrients.

Other nutrients were less mentioned and may need more intervention and education. For example, very few participants discussed zinc and iodine as nutrients to be concerned about in a plant-based diet. In addition, few participants were aware of the differences in the bioavailability of calcium, iron, and zinc and did not express knowledge regarding how to optimize absorption. The participants did not report consciously or consistently seeking these nutrients; however, they are more readily available in a wide variety of foods (Table 1). Other studies have observed similar deficiencies in knowledge as it relates to bioavailability [37,38], demonstrating an important gap.

Protein seems to be the nutrient of least concern to address, despite the limited knowledge expressed regarding complete proteins. The sample reported consuming a wide variety of plant foods containing protein throughout the diet, and studies show that protein needs are often exceeded in the average American diet (including among vegans and vegetarians) [39]. The literature also suggests that protein complementing does not need to happen at the same meal [17,40] which supports focusing on the overall dietary variety.

The variation seen in nutrition knowledge suggests a need for more health practitioners with training in plant-based nutrition in rural areas. However, considering that basic healthcare is already limited in rural areas [41], access to those who specialize in plant-based nutrition may be difficult to achieve. Opportunities for continuing education in plant-based nutrition for rural health practitioners and/or expanding opportunities for nutrition counseling through telehealth services may be valuable. Appropriately trained health practitioners can help plant-based eaters determine if dietary changes or supplement use would support their overall health. The participants augmented what they learned from health professionals with information from the internet and word of mouth, which may or may not be based on current science [42]. Bolstering plant-based nutrition education provided through existing governmental and community-based programs (e.g., Supplemental Nutrition Assistance Program Education (SNAP-Ed), cooking classes) also may help bridge the current gap in nutrition support [43].

Few participants utilized specialty products that contain nutrients of concern such as hemp seeds and dulse. This may have been due to a lack of knowledge, interest, availability, or cost barriers to accessing these products. Online grocery shopping through companies with a national reach could help improve the access to plant-based foods where availability is limited locally. However, the barriers to the adoption of online grocery shopping in rural areas have been documented previously [44,45]. Thus, it may be beneficial to promote what is already available and accepted in rural areas rather than foods that may be less accessible—given the high level of acceptance.

This study is subject to several limitations. First, our approach to operationalizing a healthy plant-based diet has not been used previously, and further work is needed to confirm the utility of this definition and validate the measure prior to continued use. Second, we did not assess dietary intake, so it was not possible to evaluate the knowledge of nutrients against the actual intake. This could be explored in future research. Third, the sample was relatively homogenous, with a higher proportion of female participants, higher income individuals, and highly educated individuals. While these characteristics do not reflect the general population [25], they do align with the national profile of people with meat-reduction behaviors [4,5]. Most of the participants were recruited through co-op grocery stores and plant-based restaurants, which may not be used by all plant-based eaters who meet the fruit and vegetable recommendation in Vermont or elsewhere. However, co-op grocers and other specialty food stores are commonly used food retailers in Vermont [46], perhaps reflecting the state’s unique food system. Further, the use of sampling quotas supported more diverse representation with respect to age, location, and gender. Population-based research on rural plant-based eaters would help confirm whether the participants in this study reflect rural residents with high-quality plant-based diets more broadly. It is important to note that any bias introduced by the targeted recruitment on positive deviants would have biased results in the direction of more nutrition knowledge rather than less. Fourth, Vermont has a unique and progressive food system, so the participants’ perspectives may not reflect those of rural plant-based eaters in other parts of the country. However, the state has long been an innovator in food-based trends in the U.S. [47,48] and it is possible that the current experiences in Vermont may be a precursor for what is to come in other parts of the country. Finally, this study included individuals with various types of plant-based diets. Further inquiries could be performed on single diet types to gain more in-depth knowledge about each group.

## 5. Conclusions

Diets rich in minimally processed plant-based foods have been associated with a reduced risk of selected non-communicable diseases; however, limiting animal-source foods may create difficulties in meeting the recommendations for some key nutrients. Inadequately planned plant-based diets may be especially problematic for those living in rural areas where the access to qualified practitioners familiar with plant-based nutrition may be more limited and environmental, economic, and cultural barriers to eating a nutritious plant-based diet have previously been identified. This study found important nutrition knowledge gaps among a sample of rural plant-based eaters related to calcium, iron, zinc and iodine, and to a lesser degree Vitamin B12, Vitamin D, omega-3s, and protein.

These findings suggest the need for action to improve the knowledge of and access to the nutrients of concern among rural residents who limit their intake of meat and other animal-source foods. Potential interventions include building the capacity of rural healthcare providers to offer relevant nutrition support and action in the food supply to make nutrient-rich plant-based foods and supplements more readily available and affordable to all. Rural communities may be well positioned to leverage their agricultural resources to emphasize the local agriculture, retail, and food services.

Future studies are needed to explore these topics in larger, representative populations of rural U.S. adults, investigate the associations between knowledge and nutrition status among plant-based eaters, and identify the information and training needs related to plant-based diets among rural healthcare providers.

## Figures and Tables

**Table 1 nutrients-16-03504-t001:** Overview of nutrients of concern in a plant-based diet.

Nutrient	Key Sources	Key Roles in the Body	Limitations of a Plant-Based Diet
Vitamin B12	Eggs, Dairy, Meat, Nutritional Yeast	DNA Synthesis, Nerve Function, Red Blood Cell Formation, Energy Metabolism,	Not present in sufficient quantities in plant foods
Vitamin D	Fatty Fish, Eggs, Fortified Milk, Mushrooms, Exposure to sunlight	Bone Health, Immunity	Not present in sufficient quantities in plant foods
Omega-3 Fatty Acids	Fatty Fish, Flax Seed, Chia Seed, Walnuts	Eye Health, Heart Health, Brain Health	Plant foods only contain ALA, which is not efficiently converted to DHA/EPA
Calcium	Dairy, Dark Green Leafy Vegetables	Bone Health, Muscle Function, Nerve Transmission, Blood Clotting	Absorption may be impacted by low Vitamin D
Iron	Red Meat (Heme), Enriched Grains, Spinach, Legumes (non-Heme)	Oxygen Transport	Non-heme iron is not as easily absorbed
Iodine	Iodized Salt, Seafood, Seaweed, Dairy	Thyroid Health	Present in lower quantities in plant foods
Zinc	Meat, Seafood, Nuts, Legumes, Grains	Immunity, Cell Growth and Development, DNA Synthesis	Absorption inhibited by phytates in plants
Protein	Meat, Eggs, Soy, Nuts, Seeds, Legumes	Building blocks for muscle, bones, hormones, and enzymes	Many plant proteins do not contain all the essential amino acids

**Table 2 nutrients-16-03504-t002:** Sociodemographic characteristics of the participants.

	*n*	Percent ^1^
Age		
Under 50 years old	13	46.4
Over 50 years old	15	53.6
Sex		
Female	21	75
Male	7	25
Education		
<Bachelor’s Degree	5	17.9
Bachelor’s Degree	13	46.4
Graduate Degree	9	32.1
Income		
Less than $24,999	6	21.4
$25,000–$74,999	9	32.1
$75,000–$99,999	7	25
$100,000 or greater	6	21.4
Race/Ethnicity		
Non-Hispanic White	22	78.6
BIPOC	3	10.7
Did not disclose	3	10.7
Rurality ^2^		
Counties in a metro area of fewer than 250,000 population	6	21.4
Nonmetro counties with an urban population of 20,000 or more, adjacent to a metro area	11	39.3
Nonmetro counties with an urban population of 2500 to 19,999, adjacent to a metro area	5	17.9
Nonmetro counties with an urban population of 2500 to 19,999, not adjacent to a metro area	6	21.4
Diet Type		
Flexitarian	6	21.4
Pescatarian	6	21.4
Vegan	11	39.3
Vegetarian	5	17.9

^1^ Due to rounding, not all numbers sum to 100. ^2^ Based on USDA Rural-Urban Continuum Code (RUCC) metrics.

**Table 3 nutrients-16-03504-t003:** Summary of the key insights and illustrative quotes for each of the nutrients queried.

Nutrient	Key Insights	Illustrative Quotes
Vitamin B12	- Overall high awareness of the limitations of a plant-based diet - Highest concern among vegans - Commonly supplemented - Use of nutritional yeast for both supplementation and flavor	*[I take] B12 is because that’s a vitamin that you get basically from animal products…. So being a vegetarian, I just don’t get that.*—Vegan, Female, 64, P#15 *I’d say the B12 is the only thing I kind of take like with a dietary mindset…the only thing I can think of that I eat that has B12 in it is like B12 fortified nutritional yeast…*—Vegan, Male, 38, P#20
Vitamin D	- Concern was more related to limited food sources and living at a northern latitude, than to a plant-based diet - Supplementation commonly recommended by healthcare practitioners	*I don’t know what foods are great in Vitamin D… [and] I don’t always get out a lot, so like I just started taking [a Vitamin D supplement] for peace of mind*.—Vegan, Male, 38, P#20 *If you’re not getting [Vitamin D] from dairy, which is fortified with Vitamin D, then then you gotta get it from a supplement. So we take a vitamin B3 and a Vitamin D*.—Vegan, Male, 32, P#25
Omega-3 Fatty Acids	- Knowledge and concern most common among vegans - Limited knowledge about the distinction between ALA and EPA/DHA - Use of algal-based DHA/EPA omega-3 supplements among a small sub-set - Nuts and seeds commonly consumed, but not necessarily as a conscious ALA omega-3 source	“*…with the Omega fatty acids our bodies just need that. And…like most things that if you’re just eating a vegan diet and not thinking about it, you’re just not gonna get that probably. And so again I think that’s something to like seek out or make sure you’re having at least like a certain level*.”—Vegan, Male, 44, P#24 “*[Our Omega supplement] comes from algae oil, which is the…basis of how fish get it…*”—Vegan, Male, 32, P#25
Calcium	- Focused on food sources as opposed to supplementation - Concerned more about the context of other health conditions such as bone health and aging, but not exclusively plant-based diets - Some believe that dairy is needed for adequate calcium - Less concern among non-vegans - Limited knowledge about the decreased bioavailability of calcium in plant-based foods or role of Vitamin D in absorption	*Usually I… dry and powder greens, and then I put that in everything. So to try to keep the calcium because there’s a lot of calcium in dark greens.*—Vegan, Female, 52, P#11 *Hopefully if you’re eating dairy, you’re going to get to plenty of calcium. But the bioavailability of calcium and iron in a lot of plant-based foods can be really limiting. So just being conscious of that.*—Flexitarian, Female, 61, P#01
Iron	- Across all diet types, concern is not necessarily related to a plant-based diet and is more relevant for those who menstruate - Focus on iron-rich foods as opposed to supplementation - Belief that red meat is not needed for adequate iron - Limited knowledge about the decreased bioavailability of plant-based sources	*…I’m not sure I’m getting enough of this iron…they don’t let me donate blood because my iron’s too low….But that’s a common problem for menstruating people anyway, so it might be unrelated to the diet.*—Flexitarian, Female, 33, P#12 *Because we select from all different types of veggies… I don’t worry about [iron] too much, but I do try to stick with some of the green leafys to keep my iron up.*—Vegan, Female, 51, P#02
Iodine	- Minimal concern in the context of a plant-based diet across all diet types - Seaweed/dulse was a popular food for both nutritional and taste purposes - Preference for seaweed as a source of iodine over iodized salt	*I make sure I have at least a pinch of dulse every day to try and make sure there’s enough iodine because that is a quick deficiency you can run into if you have it, yeah.*—Vegan, Female, 64, P#05 *I don’t use, just like iodized salt a lot. Like I even use like a substitute like seaweed more than that probably.*—Vegan, Male, 44, P#24
Zinc	- Limited knowledge about the limitations of zinc absorption in a plant-based diet - Wide consumption of zinc-containing foods, without a mention of zinc	N/A
Protein	- Tofu, tempeh, beans, lentils, nuts, and seeds were commonly consumed plant-based protein foods - Mixed level of concern about the adequacy of protein in a plant-based diet; no concern was expressed by vegans - Moderate knowledge among vegans and vegetarians about protein complementing	*I generally follow the guidelines of some kind of grain with bean combination, because it makes it more of a complete protein… but try to keep that mindful.*—Vegan, Female, 52, P#11 *I think that [protein is] the biggest thing that people freak out about…I think we overemphasize protein as a culture. So I don’t ever worry about that and I feel strong like I’ve never felt weak…as a vegetarian*—Pescatarian, Female, 30, P#27 *[By] not eating meat I lack [protein and B12] in my diet, so I supplement those by either seeking out extra protein or like focusing on it in particular, or by taking a supplement.*—Vegetarian, Female, 20, P#06

## Data Availability

Due to the small size of the communities where we conducted the interviews, the sometimes high visibility of plant-based eaters in rural areas, and ethical considerations related to participant confidentiality, it is not possible to make data from this study publicly available.

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
