# Peer review of "A Qualitative Study of Rural Plant-Based Eaters’ Knowledge and Practices for Nutritional Adequacy"

_nutrients, 2024, doi:10.3390/nu16203504_

Round 1
Reviewer 1 Report
Comments and Suggestions for Authors
The investigation revealed a number of significant major limitation:
The operationalization of a healthy plant-based diet has not been previously defined, highlighting the need for additional validation of this definition.
One limitation of the study was the absence of an assessment of the participants' actual dietary intake. Consequently, it was not feasible to compare their knowledge of nutrients with their actual consumption.
Sample Homogeneity: The sample exhibited a notable degree of homogeneity, characterized by a greater representation of female participants, persons with higher wealth, and those with a high level of education. This demography is not indicative of the overall population and may not effectively represent the wider community of rural individuals who consume plant-based meals.
Primarily recruited through co-op grocery stores and plant-based restaurants, participants may not be representative of all plant-based eaters in Vermont, therefore introducing a potential targeted recruitment bias. Implementing this focused recruitment strategy may add bias, resulting in findings that indicate a higher level of knowledge compared to the overall community of plant-based eaters.
Additionally Minor revision:
An inherent constraint of the study is the possible absence of uniformity in the recruitment techniques. The recruitment of participants mainly through community groups, local food co-ops, and plant-based eateries may have resulted in a sample that lacks complete representativeness of all rural plant-based consumers. Participants who do not regularly visit these establishments or who may have alternative eating habits may have been inadequately represented, therefore impacting the applicability of the results to the wider demographic of rural plant-based eaters.
In addition, the study did not evaluate the participants' real food intake, therefore restricting the capacity to establish a correlation between their knowledge of nutrients and their consumption habits. This may result in the omission of any gaps in comprehension that may exist in clinical settings.
Reviewer 2 Report
Comments and Suggestions for Authors
I am very grateful to you for the invitation to review the manuscript nutrients-3195799 by Leonetti and coauthors, titled "A qualitative study of rural plant-based eaters’ knowledge and practices for nutritional adequacy". This study conducted semi-structured interviews with rural plant-based eaters who believed themselves to have healthy diets. The objectives were to determine the level of knowledge and awareness of nutrients of concern among plant-based eaters and identify where they access nutrition information. The work is interesting but needs adjustments to improve the quality of the material.
Comments:
- Lines 15-16: Specify which components promote these benefits.
- Line 17: Specify the main nutritional deficiencies related to the removal of animal products from the diet.
- Lines 18-20: Briefly specify the importance of these selected components for evaluation.
- Lines 24-25: Review the sentence.
- Line 30: Replace repeated keywords with different terms not mentioned in the title.
- Lines 34-36: Please insert numerical data to support the statement.
- Lines 39-42: Insert data on the population size of each highlighted group.
- Lines 43-46: There needs to be at least a minimal relation between the dietary components that impact or reduce the risks of the diseases presented.
- Line 53: It is necessary to correlate these components with their benefits or functions in the human body.
- Introduction: Include how the data can be used to improve the diet of the rural population or any other relevant strategy.
- Lines 82-83: Was this definition created by the authors? This needs clarification since the number of meals involving the consumption of animal products is quite high.
- Results: Clearly present the participants' views regarding the type of diet.
- Lines 362-372: Despite its importance, this section should be used to support or challenge the interviewees' claims.
- Discussion: There is no general explanation or correlation between the evaluated components' functions and their roles in the body or impacts on health. The issues associated with not consuming these components are not described. Beyond describing the interviewees' perceptions, it is necessary to clearly explain the role of each component and the motivation for supplementation in many cases. I suggest expanding the discussion to better support the results. Additionally, consider merging the results and discussion sections to make these observations clearer.
- Discussion: The authors should address future actions or practices that can be implemented to improve access to information or correct the diet for better nutritional balance.
- Conclusion: Please remove references from the conclusion.
- Appendix: Verify whether this section is necessary or if it can be included as supplementary material.
Round 2
Reviewer 1 Report
Comments and Suggestions for Authors
No issue
Reviewer 2 Report
Comments and Suggestions for Authors
The article has been modified in accordance with the suggested corrections and has been substantially revised to align with the required standards.